# Uptake of Methylene Blue from Aqueous Solution by Pectin–Chitosan Binary Composites

**Dexu Kong** [1,2] **and Lee D. Wilson** [2,*]

1    Saskatchewan Research Council, 125-15 Innovation Boulevard, Saskatoon, SK S7N 2X8, Canada;
     dek593@mail.usask.ca
2    Department of Chemistry, University of Saskatchewan, 110 Science Place, Saskatoon, SK S7N 5C9, Canada
*    Correspondence: lee.wilson@usask.ca; Tel.: +1-306-966-2961; Fax: +1-306-966-4730

**Abstract:** To address the need to develop improved hybrid biopolymer composites, we report on the preparation of composites that contain chitosan and pectin biopolymers with tunable adsorption properties. Binary biopolymer composites were prepared at variable pectin–chitosan composition in a solvent directed synthesis, dimethyl sulfoxide (DMSO) versus water. The materials were characterized using complementary methods (infrared spectroscopy, thermal gravimetric analysis, pH at the point-of-zero charge, and dye-based adsorption isotherms). Pectin and chitosan composites prepared in DMSO yielded a covalent biopolymer framework (CBF), whereas a polyelectrolyte complex (PEC) was formed in water. The materials characterization provided support that cross-linking occurs between amine groups of chitosan and the –COOH groups of pectin. CBF-based composites had a greater uptake of methylene blue (MB) dye over the PEC-based composites. Composites prepared in DMSO were inferred to have secondary adsorption sites for enhanced MB uptake, as evidenced by a monolayer uptake capacity that exceeded the pectin–chitosan PECs by 1.5-fold. This work provides insight on the role of solvent-dependent cross-linking of pectin and chitosan biopolymers. Sonication-assisted reactions in DMSO favor CBFs, while cross-linking in water yields PECs. Herein, composites with tunable structures and variable physicochemical properties are demonstrated by their unique dye adsorption properties in aqueous media.

**Keywords:** composites; chitosan–pectin; adsorption; polyelectrolyte complex; covalent biopolymer framework

## 1. Introduction

Methylene blue is a cationic dye with high water solubility that has many diverse applications which includes the dyeing of paper, cotton, wool, and hair [1]. The occurrence of synthetic dyes in industrial effluent has led to global environmental concern due to the inadvertent release into aquatic environments and the impacts of such contaminants on ecosystems and human health. Exposure to lethal doses of cationic dyes such as methylene blue may lead to vomiting, cyanosis, jaundice, shock, and tissue necrosis in humans [2]. To address the removal of synthetic dyes from industrial wastewater, conventional methods such as electrochemical, coagulation, flocculation, chemical oxidation, solvent extraction, and adsorption have been reported [3–5]. Among these methods, adsorption is a popular choice for contaminant removal due to its simplicity of operation, cost-effectiveness, and availability of commercial adsorbents such as zeolites and activated carbon. Whereas the efficiency of adsorption processes is often limited by the physicochemical properties of the adsorbent and its regeneration capability, there is continued interest that exists in the development of biomaterial adsorbents derived from renewable sources such as cellulose and chitosan. Chitosan is a natural product derived from chitin upon deacetylation via alkaline hydrolysis, where the resulting copolymer contains glucosamine

and acetylated glucosamine co-monomer units. The solubility of chitosan and its chemical reactivity scale as the degree of deacetylation reaches 60% or more [1]. The synthetic versatility of chitosan is evidenced by its various modified forms upon surface functionalization, cross-linking, and composite formation. As well, the physical modification of native chitosan through the alteration of its morphology in the form of nanomaterials, beads, and fibers can also lead to changes in the textural and adsorption properties toward ionic species. Raw chitosan and its derivatives are promising biopolymers for cation–anion adsorbate binding due to its unique adsorption properties [1,6,7].

The continued interest in the development of biopolymer-based sorbents provides an opportunity to develop sustainable adsorbent technology [8]. A previous study by Sabzevari et al. on the preparation of chitosan composites that contain graphene oxide (GO) displayed unique adsorption with methylene blue, as compared with pristine chitosan. Whereas GO is an arene base fragment with polar functional groups (-OH, -COOH) due to the controlled oxidation of graphite, pectin contains $\alpha$-(1–4) linked D-galacturonic acid units and $\alpha$-(1–2) linked L-rhamnopyranose residues. In comparison to GO, pectin is a suitable precursor for the preparation of cross-linked chitosan-based composites [9,10] due to its relatively low $pK_a$ ($pK_a$ = 2.9–3.2). The galacturonic acid (GalA) groups of pectin can react with methanol in an acidic environment to form methyl esters, where the majority of these GalA units are present as methyl esters in their native form. The degree of substitution (DS) of methyl ester formation is used to classify pectin polymers, where such biopolymers with a high methyl ester content (DS > 50%) are referred to as HM pectins [11].

Since chitosan contains glucosamine and N-acetyl glucosamine units linked through a (1–4) linkage, the biopolymer can exist in its cationic form upon protonation at acidic pH below its $pK_a$. The protonated amine groups of chitosan are considered as the active sites to attract anion species through electrostatic interactions. As well, the amine groups can also undergo reaction with carboxylic acids to form hybrid composite materials [9,12]. Chitosan-based composites that contain pectin may undergo covalent or ionic bonding, as shown by the formation of a polyelectrolyte complex (PEC) or amide linkages between chitosan and pectin to yield a covalent biopolymer framework (CBF), as conceptually illustrated in Scheme 1.

In this study, various pectin–chitosan composite adsorbents were prepared and their physicochemical properties were characterized using infrared (IR) spectroscopy and other complementary methods. The dye adsorption properties of the composites were studied using methylene blue (MB), which is a model cationic dye that can provide insight on the nature of composites formed between pectin and chitosan (cf. Scheme 1). The overall goal of this study was to synthesize and characterize novel hybrid biopolymer adsorbents derived from chitosan and pectin, where the following objectives were addressed: (1) to synthesize pectin–chitosan composites at variable composition ratios using two different solvents (DMSO versus water), (2) to characterize the structure and physicochemical properties of the composites using complementary methods, and (3) to characterize the equilibrium adsorption properties of the biopolymer composites using methylene blue as a dye probe. This research addresses the knowledge gap concerning the structure–adsorption properties of pectin and chitosan composites according to the mode of preparation.

**Scheme 1.** Formation of pectin–chitosan composites through a polyelectrolyte complex (PEC) or a covalent biopolymer framework (CBF).

## 2. Materials and Methods

### 2.1. Materials and Equipment

Chitosan (Mwt. approximately 50,000–190,000 g/mol) with an average deacetylation of 75–85%, dimethyl sulfoxide (> 99.7%, DMSO), methylene blue (MB), and pectin from citrus peel galacturonic acid ≥ 74.0% (dry basis) were obtained from Sigma-Aldrich (Edmonton, AB, Canada). Ultrasonic homogenizer (Fisherbrand[TM] Model 505) is from Fisher Scientific (Edmonton, AB, Canada).

### 2.2. Synthesis of Pectin and Chitosan Composites

#### 2.2.1. Pectin–Chitosan Polyelectrolyte Complexes in Water: PC15 W, PC11 W, PC51 W

To prepare the 1:5 pectin/chitosan composite (PC15 W), the 2 wt % chitosan solution was prepared by dissolving chitosan (ca. 2 g) into 98 g of acetic acid (2 wt %), and the pectin solution was prepared by dissolving of pectin (ca. 1.37 g) into 68.6 g of deionized water to make a 2 wt % solution. In a 150 mL beaker, 50 g of the 2 wt % chitosan solution was mixed with 10 g of the pectin 2 wt % solution at 23 °C with a magnetic stirrer at 1000 rpm overnight. The mixture was neutralized with 1M NaOH (*aq*) 12 h after the mixing step until pH 7 to yield a suspension of pectin–chitosan particles. The pectin–chitosan composite precipitate was filtered by a vacuum pump with Whatman 42 ashless filter paper and

washed with deionized water, where the filtrate reached a low conductivity (35 μS/cm). The final products were air-dried for 48 h. Procedures for making PC11 W and PC51 W were similar to the previous procedures to obtain the PC15 W products, except the relative amount of pectin and chitosan was varied to achieve the different weight ratios accordingly.

### 2.2.2. Sonication-Assisted Synthesis of Pectin–Chitosan Composites in DMSO: PC15 S DMSO, PC11 S DMSO, and PC51 S DMSO

To prepare PC15 S DMSO, pectin (ca. 1.5 g) and chitosan (ca. 7.5 g) were dispersed in DMSO (200 mL). The pectin and chitosan mixture in DMSO was sonicated for 10 min. After cooling, the brown-dark pectin/chitosan composites were filtered and dried in the fume hood at 23 °C. The preparation of the PC11 S DMSO and PC51 S DMSO composites were similar to the above except that pectin (ca. 2 g and 7.5 g) and chitosan (ca. 2 g and 1.5 g) were suspended in DMSO (200 mL).

### *2.3. Characterization of Composite Materials*

The characterization of the composites involved the use of several complementary methods: pH at the point-of-zero charge ($pH_{pzc}$), Fourier Transform Infrared (FTIR) spectroscopy, ultraviolet-visible (UV-vis) spectrophotometry, and thermal gravimetric analysis (TGA).

### 2.3.1. pH at the Point-of-Zero Charge ($pH_{pzc}$)

The relative surface charge of the sample was determined by estimation of the $pH_{pzc}$ using an adapted method reported previously [13]. First, 0.01 M NaCl aqueous solution (20 mL) was transferred into each of nine 7-dram glass vials. The solution pH of each vial was adjusted by the addition of NaOH/HCl to obtain pH values that ranged from pH 2.0 to 13.0. The sample materials (ca. 0.5 g) were added to each solution and allowed to equilibrate for 48 h before the final pH was measured. A graph of final pH versus initial pH was plotted where the intersection point was recorded as the pH for the point-of-zero charge ($pH_{pzc}$) for each material.

### 2.3.2. FTIR Spectroscopy

Fourier transform infrared (FTIR) spectra of powdered samples were obtained as 1 wt % solid samples mixed with KBr and analyzed in diffuse reflectance mode using a BIO-RAD FTS-40 spectrophotometer. Multiple ($n = 64$) scans were obtained with a 4 cm$^{-1}$ resolution that was corrected against a background spectrum of spectroscopic grade KBr over a defined spectral range (400−4000 cm$^{-1}$).

### 2.3.3. Thermal Gravimetric Analysis (TGA)

TGA profiles were obtained using open aluminum pans with a TA Q50 (New Castle, DE) instrument. The heating rate (5 °C min$^{-1}$) profile was monitored from 30 to 500 °C using a $N_2$ purge gas environment.

### 2.3.4. Dye Adsorption Studies

The adsorption properties of the samples were evaluated using methylene blue (MB) in batch mode. Stock aqueous solutions of MB (0.2–10 mM) were prepared at pH 6 under ambient conditions. For each different set of 3-dram glass vials, adsorbents (ca. 10 mg) were added along with the MB solution (10 mL) at variable dye concentration (0.2–10 mM). The vials were sealed with parafilm and mixed in a horizontal shaker for 24 h at 130 rpm. After mixing for 24 h, the system reached equilibrium and the samples were centrifuged, where the supernatant containing MB was analyzed using UV-vis spectrophotometry. The optical absorbance of MB was determined using a Shimadzu UV-vis spectrophotometer (Bio-RAD FTS-40 IR spectrophotometer, Bio-Rad Laboratories, Inc., Philadelphia, PA, USA) at the maximum absorbance ($\lambda_{max} = 662$ nm) to yield a calibration curve across a concentration

range of dye (0.1–10 mM) [14]. The dye adsorption properties of the samples were evaluated by measuring the concentration of unbound MB in the supernatant phase.

Adsorption of Methylene Blue (MB)

The molar absorptivity of methylene blue ($\varepsilon$-$_{MB}$) was estimated by the Beer–Lambert law using linear calibration plots and compared against literature values. UV-Vis spectroscopy can be used to determine residual levels of MB in solution after the adsorption process from the experimental value of $\varepsilon$-$_{MB}$. Adsorption was carried out for all composites at ambient conditions, where the uptake of dye by the composites was determined from the difference between the initial dye concentration ($C_o$) and the residual dye concentration ($C_e$), as described by Equation (1).

$$Q_e = \frac{(C_e - C_o) \times V}{m} \tag{1}$$

$C_o$ (mmol) and $C_e$ (mmol) are defined above, where $V$ (L) is the volume of dye solution, m is the mass of the adsorbent (g), and $Q_e$ is the dye uptake by per mass of adsorbent (mmol/g) at equilibrium.

Adsorption Isotherms

Adsorption isotherms were obtained by plotting $Q_e$ versus $C_e$ (*cf.* Equation (1)) and were analyzed by fitting to a suitable isotherm model (*cf.* Equations (2)–(4)). The Langmuir model (Equation (2)) accounts for monolayer adsorption onto a homogeneous surface.

$$Q_e = \frac{Q_m K_L C_e}{1 + K_L C_e} \tag{2}$$

$Q_e$ and $C_e$ are defined as in Equation (1), whereas $Q_m$ is the maximum monolayer adsorption capacity of the dye per unit mass of adsorbent (mmol/g), and $K_L$ (L/mmol) is the Langmuir equilibrium adsorption constant. By comparison, the Freundlich model (Equation (3)) describes the possibility of multilayer adsorption onto a heterogeneous adsorbent surface.

$$Q_e = K_f C_e^{\frac{1}{n_f}} \tag{3}$$

$K_f$ is the Freundlich adsorption capacity constant and $n_f$ denotes the intensity of adsorption. The Sips adsorption model (Equation (4)) accounts for both Langmuir (when $n_s = 1$) and Freundlich behavior (when $n_s \neq 1$) under certain limiting conditions. The maximum monolayer adsorption capacity ($Q_m$, mmol/g) of the adsorbent can also be estimated. $K_s$ (L/mmol) is the Sips equilibrium adsorption constant, and $n_s$ denotes the Sips heterogeneity parameter

$$Q_e = \frac{Q_m K_s C_e^{n_s}}{1 + K_L C_e^{n_s}} \tag{4}$$

Surface Area Estimated from MB Adsorption

The dye sorption method provides an independent estimate of the adsorbent surface area (SA; m²/g), according to the following equation [15]:

$$SA = \frac{A_m Q_m L}{Y} \tag{5}$$

where $A_m$ represents the cross-sectional area occupied by MB ($A_m$, for a "coplanar" orientation is $8.72 \times 10^{-19}$ m²/mol, where the dimensions of the dye are 1.43 nm × 0.61 nm), $Q_m$ is the monolayer adsorption capacity per unit mass of sorbent, $L$ is Avogadro's number (mol$^{-1}$), and Y is the coverage factor ($Y = 2.0$ for MB) [16].

## 3. Results and Discussion

As noted above, several pectin–chitosan adsorbent materials were prepared herein according to variable synthetic conditions using adapted methods reported by other groups [17,18]. The characterization of the materials and selected physicochemical properties rely on various complementary methods: pH analysis, TGA, IR spectroscopy, and dye adsorption properties in aqueous media using methylene blue (MB). The results for the structural and physicochemical characterization of the composite materials are outlined in the sections below.

### 3.1. PZC Analysis

The point-of-zero charge (PZC) is the pH where the net surface charge of the adsorbent is zero [13]. The PZC value becomes an important parameter for interpreting interactions that occur at material surfaces, especially for charged adsorbate species when the dominant adsorption mechanism is driven by electrostatic interactions. At pH > PZC, the surface of the adsorbent shows a negative surface charge due to the adsorption of OH$^-$ ions or deprotonation of hydrogen ions. For conditions where pH < PZC, the adsorbent surface shows a positive surface charge due to the adsorption of hydrogen ions from solution [19]. In Figure 1, the PZC results show the pectin–chitosan composite that was prepared in water with a net charge of zero near pH 4.7. Since pectin is soluble in water at all pH values [20], an estimate of its pK$_a$ can be inferred according to the reported value for galacturonic acid (pK$_a$ = 3.24). An estimate of the PZC value for chitosan (ca. 6.5) has been reported [21], where changes in the PZC value upon the formation of pectin–chitosan composites reveals a unique material that differs relative to the biopolymer precursors. The reduced PZC value of the pectin–chitosan composite is within the range of an independent estimate (PZC = 4.4) [22]. On the other hand, the pectin–chitosan composites prepared in DMSO with sonication show a net charge of zero near pH 3.8. The lower PZC value for composites prepared in DMSO may reflect the greater contribution of the pectin fraction, according to the lower *pK$_a$* value estimated for galacturonic acid. This implies that the level of pectin grafting onto chitosan is higher and/or there are fewer available amine groups of chitosan to buffer the hydrogen ions dissociated from pectin. In the case of a dominant electrostatic interaction, the adsorption mechanism for pectin–chitosan CBF composites with a lower PZC value have greater Coulombic attraction to cation species (MB). This is in contrast to composites with a higher PZC value that possess a reduced surface charge.

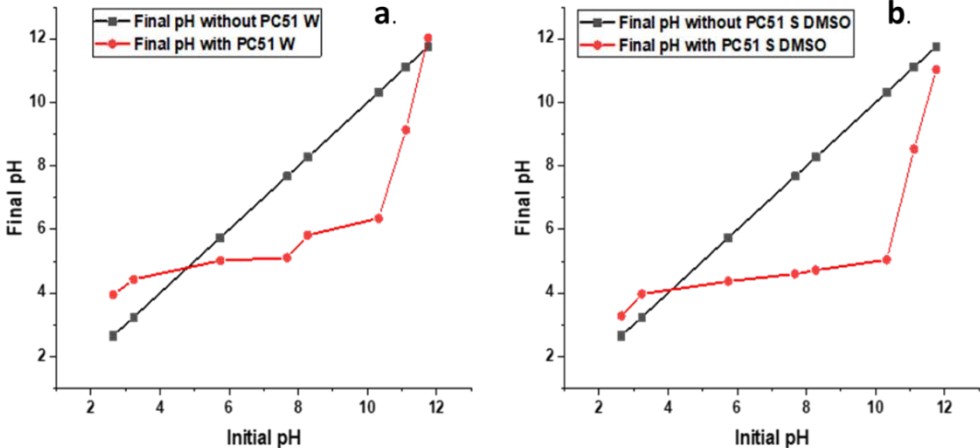

**Figure 1.** Point-of-zero charge (PZC) results of pectin and chitosan composites: (**a**) water-based synthesis; (**b**) dimethyl sulfoxide-based synthesis.

### 3.2. TGA Results of Pectin and the Pectin-Chitosan Composites

The TGA results reveal that pectin decomposes at 236 °C, which is in agreement with the results reported by Cao [23], where the main thermal event was attributed to scission of the saccharide

rings. Pal and Kaityar reported the synthesis of a lactic acid oligomer-grafted chitosan that undergoes decomposition at a lower onset temperature (ca. 200 °C) relative to that of pristine chitosan (ca. 200 °C) [24]. By comparison, the pectin–chitosan composite (PC11 S DMSO) prepared under sonication-assisted synthesis in DMSO reveals a higher decomposition temperature in Figure 2, as compared with the composites prepared in water (PC11 W and PC51 W). The greater decomposition temperature of the composites formed under sonication in DMSO (PC51 S DMSO, PC11 S DMSO, and PC15 S DMSO) provides support of the different bonding that results from water- versus DMSO-based composite syntheses. Materials prepared in water are anticipated to favor the formation of PECs due to the higher dielectric constant of aqueous media. Composites synthesized in DMSO are more likely to form CBFs due to amide bond formation between pectin and chitosan biopolymers. The FTIR results in Figure 3a for DMSO-based preparations provide confirmation of secondary amine features (two bands ca. 2900 cm$^{-1}$) that provide support for amide bond formation [25]. By contrast, pectin and chitosan composites prepared in water reveal a prominent thermal event at 220–230 °C that indicates the formation of PECs by electrostatic interactions. The TGA results for the PEC and CBF materials are in agreement with independent results from the preparation of related composites [12,26,27].

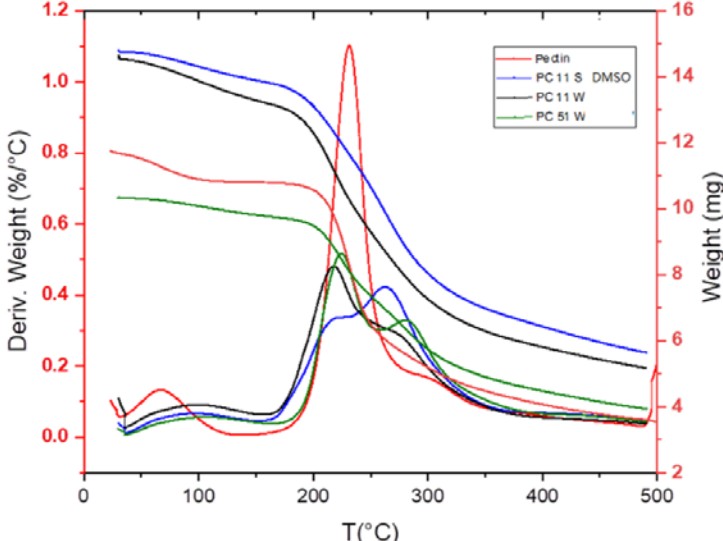

**Figure 2.** Thermal gravimetric analysis (TGA) results of pectin and pectin–chitosan composites.

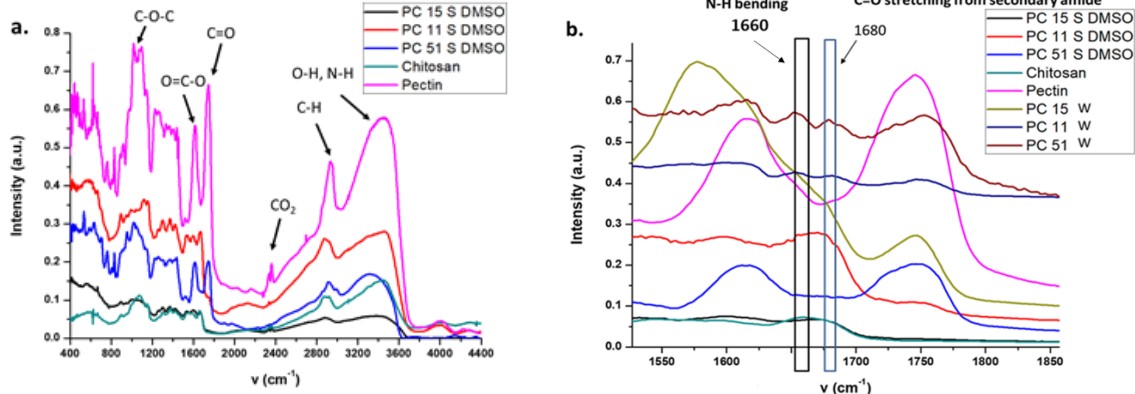

**Figure 3.** (**a**) Infrared (IR) spectra of pectin, chitosan, and their composites prepared in DMSO under assisted sonication, and (**b**) IR spectra of pectin, chitosan, and composites prepared in water and DMSO solvents.

### 3.3. FTIR Spectral Results

In Figure 3a, pectin reveals a strong intensity stretching band (C=O) from non-ionized carboxy groups (-COOH and -COOCH$_3$) of galacturonic acid at 1750 cm$^{-1}$, and lower intensity bands for the symmetric and anti-symmetric carboxylate (-COO$^-$) vibration at 1442 and 1673 cm$^{-1}$, which concur with spectral results for pectin from other reports [28,29]. The broad IR band at 1600 cm$^{-1}$ for the chitosan spectrum relates to the N–H bending of a primary amine group of the glucosamine units. The shift of this N–H band to 1660 cm$^{-1}$ for the pectin–chitosan composite indicates a change in the chemical environment of this group upon interaction with pectin. The IR shift results are consistent with the reported results of such pectin–chitosan PECs [12,26,28]. Various reported studies of pectin–chitosan composites indicate the formation of PECs in water [18,30], which are supported by the IR results for composites prepared in water in Figure 3b. In Figure 3b, the IR bands for the pectin polymer at 2924 cm$^{-1}$ indicate C–H stretching vibrations, and IR signatures between 950 cm$^{-1}$ and 1200 cm$^{-1}$ relate to the IR absorption of the pyranose ring of pectin [31]. The C=O stretching band of galacturonic acid from the pectin–chitosan composites (prepared in DMSO) appear at 1750 cm$^{-1}$ and relate to non-ionized carboxy groups (-COOH and -COOCH$_3$). The bands at 1442 cm$^{-1}$ and 1637 cm$^{-1}$ are assigned to the symmetric and anti-symmetric vibration of carboxylate (-COO$^-$) groups [31]. By comparison, the carboxylate band intensity increased, whereas the band intensity of non-ionized carboxy groups decreased, according to the formation of PECs between chitosan and pectin [31] A comparison of the IR signatures of pectin–chitosan systems in water (PECs) with composites prepared in DMSO through sonication reveal an increased intensity in the C=O stretching from a secondary amide that provide support for amide bond formation between pectin and chitosan [24,26,27,29,32]. Notwithstanding the difference in solvent properties, a rationale for the product distribution between the water and DMSO synthesis can be attributed to the modes of energy employed. Sonication-assisted synthesis differs from conventional heating and stirring, since ultrasonic waves can create vapor cavities around the surface of dispersed solids due to heating and subsequent pressure gradients due to cavitation effects. The resulting temperature and pressure gradients adjacent to the reactant surface can facilitate the amide bond formation [33]. Udoetok et al. [27] reported enhanced cross-linking effects at ambient temperature conditions in the case of epichlorohydrin cross-linked cellulose. The formation of amide linkages between pectin and chitosan is further supported by the increased signature of amide II band (N–H) bending of NH$_2$ from chitosan at 1595 cm$^{-1}$ for DMSO-based composites. The IR results provided herein are also supported by other reported studies of amide bond formation for related chitosan composite materials [9,24,26,27,29].

### 3.4. Sorption Isotherm Results

Dye adsorption isotherm results have been shown to provide insight on structurally similar systems due to the sensitivity of dye probe to its chemical environment, especially dyes with large molar absorptivity values. The change in dye adsorption reveals the variable surface accessibility of active sites on the adsorbent material due to differences in morphology and the number of active adsorption sites [34]. The trend in dye adsorption for MB with the various composites prepared in water and DMSO are shown in Figure 4, along with a comparison with results for pristine pectin. In all cases, the isotherm profiles show a nonlinear increase in dye uptake with increasing MB concentration. In the case of composites, the dye adsorption capacity increases as the pectin content increases, where the composites prepared in DMSO show notably greater uptake versus the composites prepared in water. The observed trend parallels the greater negative surface charge of composites prepared in DMSO versus the products prepared in water, which are in agreement with the offset in PZC values for each synthetic preparation. The uptake of MB by pectin and pectin–chitosan composites in aqueous solution were analyzed by several adsorption isotherms. According to Figure 4, the best-fit isotherm results for the adsorption profiles of pectin and pectin–chitosan composites with MB dye were obtained using the Sips model. Table 1 shows the Sips isotherm parameters, where K$_s$ is the Sips adsorption constant that relates to the adsorption energy, $Q_m$ is the monolayer adsorption capacity of MB, and n$_s$ indicates the

adsorbent surface heterogeneity [6]. The $Q_m$ values for the composites reveal an incremental uptake of MB as the weight fraction of pectin increased. The values of $Q_m$ (mmol/g) listed in Table 1 reveal that pectin has the greatest MB uptake capacity, which concurs with its abundant –OH and –COOH active sites. As well, pectin is very soluble in water with highly accessible carboxylate groups since pH > $pK_a$, which is in contrast to heterogeneous adsorbents that are water-insoluble with lesser surface accessibility [35]. The formation of pectin–chitosan composites with covalent amide bonding show promising dye uptake performance such as PC51 S DMSO, since such CBF-based systems are more amenable to phase separation and recovery after the dye adsorption process. Insoluble composites are contrasted with pristine pectin, in spite of the relatively high adsorption capacity of pectin. In the case of PECs prepared in water such as PC51 W, lower dye uptake is observed relative to PC51 S DMSO. The enhanced adsorption of MB by the PC51 S DMSO system can be attributed to its relatively high pectin content and the branched structure of this CBF-based adsorbent. The covalent framework likely contributes to potential cooperative effects between the biopolymer subunits to afford secondary adsorption sites for MB along the chitosan backbone. The primary adsorption sites are attributed to the carboxylate groups of pectin due to the key role of electrostatic interactions with MB. The prominent role of the carboxylate sites is evidenced by the unitary exponential term ($n_s \approx 1$) for the composites in Table 1, irrespective of the composition of the biopolymer composite. Hence, the use of MB as a dye probe enables elucidation of the active adsorption sites (–COOH, –OH, and –NHR) for pectin and the pectin–chitosan composites.

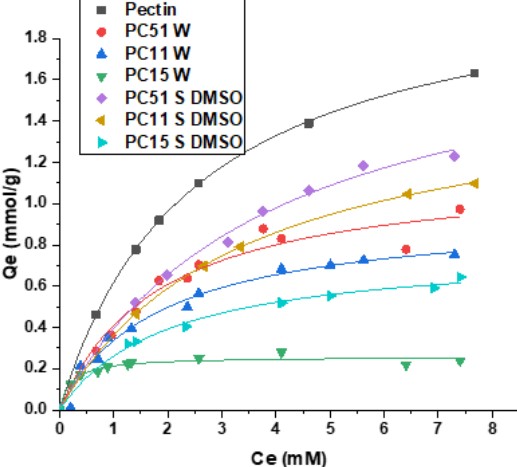

**Figure 4.** Methylene blue (MB) dye uptake isotherms for pectin and pectin–chitosan composites, where the solid lines represent best-fit results by the Sips isotherm model.

**Table 1.** Sips model fitting parameters for methylene blue (MB) dye uptake by pectin and pectin–chitosan composite adsorbents.

| Sample Name | $K_s$ (M$^{-1}$) | $Q_m$ (mmol/g) | $n_s$ | Adjusted R$^2$ | SA (m$^2$/g) [1] |
|---|---|---|---|---|---|
| Pectin | 0.50 ± 0.015 | 2.2 ± 0.012 | 0.99 | 0.99 | 170 |
| PC51 W | 0.54 ± 0.10 | 1.2 ± 0.089 | 1.3 | 0.96 | 92 |
| PC11 W | 0.54 ± 0.081 | 0.96 ± 0.051 | 1.0 | 0.97 | 75 |
| PC15 W | 0.69 ± 0.093 | 0.26 ± 0.011 | 1.1 | 0.96 | 20 |
| PC51 S DMSO | 0.36 ± 0.071 | 1.9 ± 0.10 | 1.1 | 0.97 | 146 |
| PC11 S DMSO | 0.45 ± 0.023 | 1.6 ± 0.011 | 1.0 | 0.97 | 123 |
| PC15 S DMSO | 0.50 ± 0.0.53 | 0.78 ± 0.028 | 0.84 | 0.97 | 60 |

[1] The adsorbent surface area (SA) was estimated using an equilibrium dye adsorption method, as further described in [1].

## 4. Conclusions

The preparation of a composite with a covalent biopolymer framework (CBF) was achieved by the formation of amide linkages between pectin and chitosan using a sonication-assisted synthesis in DMSO. By contrast, the use of water as a solvent with conventional heating yielded pectin–chitosan polyelectrolyte complexes (PECs). Characterization of the covalent and ionic types of pectin–chitosan composites was supported by TGA results that revealed a more thermally stable cross-linked composite with a covalent framework (PC11 S DMSO) over the pectin–chitosan polyelectrolyte complex (PC11 W) prepared in water. The IR intensity changes for the secondary amine groups of chitosan before and after composite formation provided support for the two types of composites (CBFs and PECs). Amide-based CBFs prevail in DMSO, while PECs are favored in water-based reactions. The MB uptake capacity for the pectin–chitosan CBFs exceed that of the PECs due to the key role of the carboxylate anions of pectin. The greater dye uptake capacity of pectin highlights the prominent role of the carboxylate anion site accessibility in the CBF composites versus the PEC-based materials. The formation of an amide-based covalent network results in greater pectin incorporation onto chitosan with secondary adsorption sites along the chitosan backbone. The use of a solvent to bias the formation of CBFs versus PECs in the case of pectin–chitosan composites will contribute to the rational design of materials with improved properties for diverse adsorption-based applications. This includes solid phase extraction and recovery processes due to enhanced phase separation and the removal efficiency of waterborne species that possess positive electrostatic potential.

**Author Contributions:** Conceptualization, L.D.W.; methodology, L.D.W. and D.K.; software, D.K.; validation, L.D.W. and D.K.; formal analysis, D.K.; investigation, L.D.W. and D.K.; resources, L.D.W.; data curation, L.D.W.; writing—original draft preparation, D.K.; writing—review and editing, L.D.W. and D.K.; visualization, D.K.; supervision, L.D.W.; project administration, L.D.W.; funding acquisition, L.D.W. All authors have read and agreed to the published version of the manuscript.

**Funding:** Lee D. Wilson acknowledges the support granted by the Government of Canada through the Natural Sciences and Engineering Research Council of Canada (Discovery Grant Number: RGPIN 2016-06197).

**Acknowledgments:** The authors acknowledge the support provided by the Saskatchewan Research Council and the University of Saskatchewan.

**Conflicts of Interest:** The authors declare no conflict of interest.

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
