# Peer review of "Uptake of Methylene Blue from Aqueous Solution by Pectin–Chitosan Binary Composites"

_jcs, doi:10.3390/jcs4030095_

Round 1
Reviewer 1 Report
This manuscript “Uptake of Methylene Blue from Aqueous Solution by Pectin-Chitosan Binary Composites” (jcs-860855) presented interesting studies that might be interested to scientists and communities that working in the area of biocomposites for adsorbent application.
However, there are some issues in this manuscript that need to be addressed before publication.
Please refer below for my comments:
This manuscript contains a number of mistakes.
For example, Page 4, line 108, It should be 2.2.2.
Line 97, unit used is not consistent, “2% wt.” and sometimes used “2 wt.%”
Page 8, line 230, “Figure” used in text and sometimes used “Fig.” in text
Please proofread adequately for consistency.
Technical section
The claiming of the formation of cross-linked structures from the blends with a strong covalent framework is questionable. What is the mechanism for the crosslinking formation of the blends during solution mixing without any initiator/peroxide? Is there any heat apply during mixing?
Please provide further evidence or additional references to support the claim.
In the TGA section, Figure 2, please provide the TGA curve of chitosan as well. There are two main derivatives in the TGA curves after blending and we are not sure which decomposition peaks belong to Pectin or Chitosan after blending for PC 11 and PC 51.
Page 7, line 214, Please indicate how much increment of the decomposition temperature of the Pectin from 236 oC after blending with chitosan and forming a crosslinking network.
Author Response
Authors’ Response to Reviewer Comments on MS ID: jcs-860855
Reviewer #1
This manuscript “Uptake of Methylene Blue from Aqueous Solution by Pectin-Chitosan Binary Composites” (jcs-860855) presented interesting studies that might be interested to scientists and communities that working in the area of
biocomposites for adsorbent application.
However, there are some issues in this manuscript that need to be addressed before publication.
Please refer below for my comments:
This manuscript contains a number of mistakes.
For example, Page 4, line 108, It should be 2.2.2.
Author Response: Yes, Line 113 should change to 2.2.2
Line 97, unit used is not consistent, “2% wt.” and sometimes used “2 wt.%”
Author Response: Yes, Line 101 was changed to 2 wt.%
Page 8, line 230, “Figure” used in text and sometimes used “Fig.” in text
Author Response: Yes. All related terms were changed to Figure (from Fig.) in the revised MS.
Please proofread adequately for consistency.
Technical section
The claiming of the formation of cross-linked structures from the blends with a strong covalent framework is questionable. What is the mechanism for the crosslinking formation of the blends during solution mixing without any initiator/peroxide? Is there any heat apply during mixing?
Please provide further evidence or additional references to support the claim.
Author Response: There is a precedence for the formation of amide linkages upon heating of polyelectroyte complexes, as reported in the 1950’s (see the following: Packer, J. & Vaughan, J. (1958). In A Modern Approach to Organic Chemistry. Oxford University Press, London,pp. 256-60). More recently, it has been demonstrated that amide bonds can be formed upon heating/ageing of chitosan-based PECs (see the following: Carbohydrate Polymers 1994, 23, 211-219). Since heat is generated during the sonication process, due to cavitation, along with the use of DMSO, the formation of amide linkages without catalyst can be rationalized (see also a more recent preparation: DOI: 10.1021/acs.biomac.6b00619). Additional literature has been added to the revised manuscript to support the results presented in the revised manuscript.
In the TGA section, Figure 2, please provide the TGA curve of chitosan as well. There are two main derivatives in the TGA curves after blending and we are not sure which decomposition peaks belong to Pectin or Chitosan after blending for PC 11 and PC 51.
Author Response: The following reference was added “Carbohydrate Polymers 205, 559–564, 2019” to show the chitosan TGA results for discussion. Chitosan decomposes near 300 deg. C (a reference has been added with this information in the revised manuscript: see doi:10.1016/j.colsurfb.2009.05.020).
Page 7, line 214, Please indicate how much increment of the decomposition temperature of the Pectin from 236 oC after blending with chitosan and forming a crosslinking network.
Author Response: The heating rate was added to the revised manuscript, as follows: “the heating rate is constant at 5 °C min−1”
In summary, the authors acknowledge the constructive and insightful comments of Reviewer #1. We have further edited the manuscript for clarity, language, and syntax throughout to meet the high standards of this journal.

Reviewer 2 Report
Present manuscript entitled “Uptake of Methylene Blue from Aqueous Solution by Pectin-Chitosan Binary Composites” Overall the structure of this study is good and of benefit to larger community. I recommend minor revision for this manuscript before it could be published in the J. Compos. Sci.
Comments:
- Authors should check typo errors/sentences line 43.
- What are surface area of binary composites?
- Chitosan based missing reference should be added in introduction: Carbohydrate Polymers 205, 559–564, 2019.
- What are the effects of temperature and concentration on uptake of methylene blue in aqueous solutions? Authors should provide effect temperature and concentration on uptake of methylene blue in aqueous solutions.
- How binary composites (Chitosan-Pectin) are COF materials?
Author Response
Authors’ Response to Reviewer Comments on MS ID: jcs-860855
Reviewer #2
Present manuscript entitled “Uptake of Methylene Blue from Aqueous Solution by Pectin-Chitosan Binary Composites” Overall the structure of this study is good and of benefit to larger community. I recommend minor revision for this manuscript before it could be published in the J. Compos. Sci.
Comments:
- Authors should check typo errors/sentences line 43.
Author Response: The corresponding edits were addressed in the revised manuscript.
- What are surface area of binary composites?
Author Response: The SA was calculated from the MB adsorption data according to a reported method described in ref [1]. The results were added to Table 1 and experimental description is providedin section 2.3.4.3.
|
Sample name |
SA (m2/g) |
|
Pectin |
170 |
|
PC51 W |
92 |
|
PC11 W |
75 |
|
PC15 W |
20 |
|
PC51 S DMSO |
146 |
|
PC11 S DMSO |
123 |
|
PC15 S DMSO |
60 |
- Chitosan based missing reference should be added in introduction: Carbohydrate Polymers 205, 559–564, 2019.
Author Response: The reference was added because it also shows chitosan-based biosorbents and their application for dye removal.
- What are the effects of temperature and concentration on uptake of methylene blue in aqueous solutions? Authors should provide effect temperature and concentration on uptake of methylene blue in aqueous solutions.
Author Response: We agree that variable temperature studies provide additional information about the adsorption process, however, this was outside the objectives of the present work, as indicated in the introduction section of the manuscript (see lines 80-91).
- How binary composites (Chitosan-Pectin) are COF materials?
Author Response: We agree with the reviewer and have altered the description of the materials form COF to CBF (covalent biopolymer framework). Based on the precise definition of COFs presented by O. Yaghi (http://dx.doi.org/10.1126/science.aal1585), we agree with the reviewer that COF may not be an appropriate descriptor. Hence, the use of CBF was used as an alternative choice. The formation of covalent and polyelectrolye complexes between chitosan and pectin are supported by the characterization results, along with the additional literature citations to help support the interpretation presented.
In summary, the authors acknowledge the constructive and insightful comments of Reviewer #2. We have further edited the manuscript for clarity, language, and syntax throughout to meet the high standards of this journal.
